# Thrombotic Mechanism Involving Platelet Activation, Hypercoagulability and Hypofibrinolysis in Coronavirus Disease 2019

**DOI:** 10.3390/ijms24097975

**Published:** 2023-04-28

**Authors:** Hideo Wada, Katsuya Shiraki, Hideto Shimpo, Motomu Shimaoka, Toshiaki Iba, Katsue Suzuki-Inoue

**Affiliations:** 1Department of General and Laboratory Medicine, Mie Prefectural General Medical Center, Yokkaichi 5450-132, Japan; katsuya-shiraki@mie-gmc.jp; 2Mie Prefectural General Medical Center, Yokkaichi 5450-132, Japan; hideto-shimpo@mie-gmc.jp; 3Department of Molecular Pathobiology and Cell Adhesion Biology, Mie University Graduate School of Medicine, Tsu 514-0001, Japan; motomushimaoka@gmail.com; 4Department of Emergency and Disaster Medicine, Juntendo University Graduate School of Medicine, Tokyo 113-8431, Japan; toshiiba@juntendo.ac.jp; 5Department of Clinical and Laboratory Medicine, Yamanashi Medical University, Yamanashi 409-3821, Japan; katsuei@yamanashi.ac.jp

**Keywords:** COVID-19, bacterial infection, thrombosis, platelet activation, sCLEC-2, hypofibrinolytic state

## Abstract

Coronavirus disease 2019 (COVID-19) has spread, with thrombotic complications being increasingly frequently reported. Although thrombosis is frequently complicated in septic patients, there are some differences in the thrombosis noted with COVID-19 and that noted with bacterial infections. The incidence (6–26%) of thrombosis varied among reports in patients with COVID-19; the incidences of venous thromboembolism and acute arterial thrombosis were 4.8–21.0% and 0.7–3.7%, respectively. Although disseminated intravascular coagulation (DIC) is frequently associated with bacterial infections, a few cases of DIC have been reported in association with COVID-19. Fibrin-related markers, such as D-dimer levels, are extremely high in bacterial infections, whereas soluble C-type lectin-like receptor 2 (sCLEC-2) levels are high in COVID-19, suggesting that hypercoagulable and hyperfibrinolytic states are predominant in bacterial infections, whereas hypercoagulable and hypofibrinolytic states with platelet activation are predominant in COVID-19. Marked platelet activation, hypercoagulability and hypofibrinolytic states may cause thrombosis in patients with COVID-19.

## 1. Introduction

Coronavirus disease 2019 (COVID-19) has spread worldwide from China [1,2], resulting in a pandemic [3]. It was previously reported that approximately 2% of patients with COVID-19 died, and 5–10% developed severe and life-threatening acute respiratory distress syndrome (ARDS) [4,5,6], with many more patients developing COVID-19 developing mild or moderate illness [7,8]. Following the appearance of the omicron variants of COVID-19 [9], the mortality rate was reduced, but the incidence of infections markedly increased, resulting in a relative increase in deaths. Therefore the management of complications of COVID-19 has become increasingly important.

## 2. Macrothrombotic Complications

The relationship between COVID-19 and thrombosis including venous thromboembolism (VTE) [10], such as pulmonary embolism (PE), and deep vein thrombosis (DVT) and arterial thrombosis, such as acute cerebral infarction (ACI) [11] and acute coronary syndrome (ACS) [12], has attracted attention [13]. On the other hand, many reports on thrombotic complications, such as disseminated intravascular coagulation (DIC) [14] and thrombotic microangiopathy (TMA) [15] have been previously reported in severe sepsis due to bacterial infection. A soluble C-type lectin-like receptor 2 (sCLEC-2) assay has been recently developed as a biomarker for platelet activation [16,17,18].

We herein review, based on a large number of reports, the mechanism underlying the development of thrombosis in COVID-19, which differs from that in bacterial infection.

## 3. Incidence of Macrothrombotic Complications in COVID-19

There have been many reports on macrothrombosis, such as VTE, ACS and ACI, in general, and the incidence of all thrombosis has varied substantially (6–26%) among patients with COVID-19 [19] (Table 1).

### 3.1. VTE

There have been many systematic reviews and meta-analyses concerning VTE [10,19,20,21], and the incidence of VTE, including DVT with PE and DVT, has been reported to vary in COVID-19. VTE was reportedly found more frequently in patients who were admitted to the intensive-care unit (ICU) than in those not admitted to the ICU. More than half of COVID-19 patients with PE (57.6%) lacked DVT [20], suggesting that some cases of PE might be caused by vascular injury instead of embolism. The incidence of VTE was higher when assessed according to screening or prospective studies [10] and postmortem studies [21] than in retrospective studies. These findings suggest that the incidence of VTE is high but varies depending on the incidence and severity of COVID-19, the age and race of patients, and the details of hospitalization and prophylaxis.

### 3.2. Arterial Thrombosis

The incidence of arterial thrombosis was low (0.7–3.7%) in overall patients [20,21] and 5% among ICU admissions [21]. The frequency of ACS in patients with COVID-19 was 1.0% in overall patients and 6.0–33.0% in cases of severe disease [12,22]. A review of cardiac autopsy cases of COVID-19 found that the most common comorbidities were coronary artery disease (33%) and acute ischemia (8%) [23]. A higher mortality rate among patients with COVID-19 and ST-segment elevation myocardial infarction (STEMI) was noted in comparison to previous studies, with reported concerns being late presentation due to fear of infection, delayed care time, and poor resource allocation [24].

On the neuroimaging of COVID-19 patients, especially critically ill patients, 3.4% of patients showed COVID-19-related neuroimaging findings [25,26], such as white matter abnormalities, followed by acute/subacute ischemic infarction and encephalopathy. The incidence of ACI in patients with COVID-19 is low (0.4–1.3%) [11,26,27,28]. The risk factors for ACS and ACI in patients with COVID-19 include old age, hypertension, diabetes mellitus, coronary artery disease, and severe infection [11,28]. Accurately diagnosing arterial thrombosis is difficult in COVID-19 patients with critical illnesses and there are no routine markers for ACI (such as D-dimer for VTE), which suggests that the true incidence of arterial thrombosis may be increased in COVID-19.

**Table 1 ijms-24-07975-t001:** Pooled incidence and thromboembolism in patients with COVID-19 infections.

	ICU+Non-ICU	ICU	Non-ICU	Japan
	Pooled incidence (%)			
TH	6–26 [19]	―	―	1.86 [22]
PE	7.1–16.5 [10,20,21]	19.0–24.7 [20,21]	10.5–19.0 [20,21]	0.5 [22]
DVT	12.1–20.0 [10,20,21]	28.0 [21]	―	0.7 [22]
VTE	17.0–21.0 [10,21]	4.8–31.0 [10,19,21]	1.5–46.1 [10,19]	1.2 [22]
ACI	0.4–1.3 [11,22,29,30]	―	―	0.4 [22]
ACS	1.0 [12]	6–33 [23]	―	0.1 [22]

TH, thrombosis; PE, pulmonary embolism; DVT, deep vein thrombosis; TE, thromboembolism; ACI, acute cerebral infarction; ACS. Acute coronary thrombosis, DIC, disseminated intravascular coagulation; COVID-19; coronavirus disease 2019; ICU, intensive-care unit; Reference [10] Jiménez D et al.: 48 studies with 18,093 patients; [11] Nannoni S et al.: 61 studies with 108,571 patients; [12] Zhao YH et al.: 2277 articles with 108,571 patients; [19] Cheng NM et al.: 68 studies; [20] Suh YJ et al.; 27 studies with 3342 patients; [21] Malas MB et al.: 42 studies with 8271 patients; [22] Peiris S et al.: 63 studies; [23] Roshdy A et al.: 316 cases; [26] Kim PH et al.: 17 studies with 1394; [27] Horiuchi H et al.: one questionnaire with 5807 patients; [28] Qureshi AI et al.: 8163 patients; [29] Xiao, D. et al.: systemic review. [30] Pepera, G. et al.; systemic review.

### 3.3. Mortality

The pooled mortality rate among patients with all types of thrombosis was 23%, while that among patients without any types of thrombosis was 13%. The pooled odds of mortality were 74% higher among patients who developed thrombosis than among those who did not [21]. A systematic review of reports on COVID-19 demonstrated that thrombosis increased the risk of mortality by 161% and the risk of a critical status by 190% [29]. In addition, preexisting cerebrovascular diseases (CVDs) were linked to poor outcomes and an increased risk of death in patients with COVID-19 [30].

### 3.4. After Discharge

The incidence of events in patients with COVID-19 after discharge was 1.55% for VTE, 0.45% for acute CVD, 0.49% for ACS, 0.77% for other arterial thromboses and 1.73% for major bleeding [31]. In an analysis of patients with and without COVID-19, the incidence of ACI was 1.3% in those with COVID-19 and 1.0% in those without COVID-19, suggesting that the risk of thrombosis continues after discharge and that the management of comorbidities is important for patients with COVID-19. ACI usually occurs in the presence of other cardiovascular risk factors and is associated with a twofold increase in the risk of long hospitalization or death in patients with COVID-19 [28].

### 3.5. Asia including Japan

The incidence of acute CVD in patients with COVID-19 was shown to be higher in Asia (3.1%) than in Europe (1.1%) and North America (1.1%) [11]. A questionnaire on COVID-19-related thrombosis in 6202 patients hospitalized in Japan showed that thrombotic events occurred in 1.86% of the 5807 patients with available data including symptomatic ACI (0.4%), AMI (0.1%), DVT (0.7%), PE (0.5%), and other thrombotic events (0.4%) [27], suggesting that the frequency of VTE is low in Japan due to the low incidence and severity of COVID-19 and sufficient prophylaxis with heparin.

### 3.6. Variety of Severity and Complications of Thrombosis in COVID-19

There are large differences in severity or mortality among the COVID-19 variants. Furthermore, an increased number of patients causes an increase in severe patients with COVID-19. The level of the medical system, such as bed numbers for COVID-19, quality of ICU, medical insurance and medication, can decrease mortality or thrombotic complications. Although high mortality was observed in 2019, low mortality due to COVID-19 was observed in 2023. As many factors affect mortality or complications of thrombosis in patients with COVID-19, the evaluation of thrombosis in COVID-19 should be carefully performed (Figure 1).

## 4. Microangiopathy as DIC and TMA in COVID-19 and Other Infections

### 4.1. DIC

Although the relationship between DIC and COVID-19 has sometimes been reviewed [32], few systematic reviews have been conducted and the incidence of typical DIC in patients with COVID-19 was shown to be very low [33]. However, it has been generally reported that DIC is frequently associated with patients with other infectious diseases, and the incidence of DIC in other infectious diseases suspected to be bacterial infections is 20–70% [34,35], considering that the incidence of complications with DIC is higher in patients with bacterial infections than in those with COVID-19. The outcome of DIC in septic patients is extremely low [14]. There have been many systematic reviews and studies based on big data of the effects of DIC or sepsis treatments [36,37]. In a recent report that compared COVID-19 to bacterial infection, the mortality rate was 17.0% in patients with other pneumonia, 16.7% in patients with sepsis, and 4.3% in patients with COVID-19, suggesting that the mortality rate due to sepsis is higher than that due to COVID-19 [36]. In addition, thrombosis such as VTE, ACI or ACS is not frequently detected in patients with bacterial infection [38]. There are many differences between septic DIC and COVID-19 coagulopathy. In particular, a clot waveform analysis (CWA) [39] of activated partial thromboplastin time (APTT) showed a large difference between the two diseases. Significant prolongation of the peak time and a marked reduction in the peak height of CWA-APTT were observed in patients with overt DIC [40], whereas moderate prolongation of the peak time and a significant increase in the peak height of CWA-APTT were observed in patients with COVID-19 coagulopathy [41]. Based on these findings, a markedly increased peak height suggests hypercoagulability, while a markedly decreased peak height suggests hypocoagulability (Figure 2). These differences may be caused by hypofibrinogenemia and hyperfibrinolysis in overt DIC and hypercoagulability induced by thrombin burst and hypofibrinolysis in COVID-19 coagulopathy.

### 4.2. TMA

The association with TMA in patients with COVID-19 has been reviewed [42] and several reports described TMA in patients with COVID-19 [43], with the frequency of TMA being reported to be 1.0–20% and the outcome of TMA varying but quite poor [15]. TMA involves Shiga toxin-producing Escherichia coli (STEC)-hemolytic uremic syndrome (HUS), thrombotic thrombocytopenic purpura (TTP), atypical HUS and secondary HUS [15]. Acquired TTP is caused by the inhibitor for a disintegrin-like and metalloproteinase with thrombospondin type 1 motifs 13 (ADAMTS13) and aHUS is mainly caused by an hereditary abnormality of compliment regulation.

As ADAMTS-13 activity and the complement system are not usually examined, TTP and aHUS may not usually be diagnosed in general hospitals. However, decreased ADAMTS-13 activity and elevated C5b-9 levels have been reported in patients with COVID-19 [41,44,45]. The low incidence of TMA may be due to the lack of diagnostic biomarkers for TMA in clinical use. Elevated sCLEC-2 levels suggest that critically ill patients with COVID-19 have some degree of microangiopathy [46]. A marked elevation of sCLEC-2 levels was also reported in patients with TMA [17], suggesting the marked activation of platelets in patients with COVID-19 as well as in patients with TMA. Many critically ill patients with COVID-19 are also associated with thrombocytopenia, anemia and organ failure, suggesting that these patients met the diagnostic criteria of TMA [47] and necessitating further investigation for TMA in patients with COVID-19. COVID-19 complicated with TMA is expected to increase in frequency going forward.

## 5. Biomarkers for Thrombosis in COVID-19

### 5.1. Routine Biomarkers

Although conventional PT and APTT are hemostatic markers and cannot show hypercoagulability and thrombotic risk, CWA-APTT and a small amount of TF-induced FIX activation assay (sTF/FIXa) can show hypercoagulability [39]. D-dimer values have been reported to be useful biomarkers with a high sensitivity for thrombosis in patients with COVID-19 and are correlated with the severity of COVID-19 [19,20] (Table 2). Although elevated D-dimer levels are a well-known risk factor for thrombosis, the D-dimer cutoff level is low in COVID-19 [48]. Although D-dimer is useful for the exclusion of VTE in patients with COVID-19, it may not be useful for the diagnosis of VTE in patients with COVID-19 [15,19,20]. D-dimer levels were reported to be significantly higher in patients with other pneumonia and sepsis due to bacterial infections than in patients with COVID-19, whereas there was no significant difference in D-dimer levels between patients with unidentified clinical syndrome and those with COVID-19 [14,15,46].

Platelet counts were extremely low in patients with sepsis and other pneumonia due to bacterial infection, especially DIC or pre-DIC, but only moderately low in COVID-19 patients with critical illness [14,15,46]. As multiple viruses interfere with hematopoiesis, thrombocytopenia is a common phenomenon in various viral infections including COVID-19 [49]. However, thrombocytopenia is suggested to be associated with increased platelet consumption and destruction in COVID-19. The prothrombin time (PT) and APTT were significantly prolonged in septic patients with DIC but not in patients with COVID-19 [14,15,46]. Therefore, DIC and sepsis-induced coagulopathy are generally diagnosed using a scoring system based on PT, platelet counts and fibrin-related products, such as D-dimer levels [50,51,52]. Fibrinogen levels were significantly increased in patients with COVID-19 compared with patients with sepsis. No significant differences have been noted in platelet counts, PT or APTT among the four stages of COVID-19, although platelet counts tend to be reduced in severe or critical illness [46]. Therefore, the above scoring system may not be useful for diagnosing thrombosis in patients with COVID-19, suggesting that coagulation factor abnormalities may not be significant in COVID-19.

### 5.2. Platelet Activation

Platelet activation can be evaluated to detect substances such as P selectin [53] or phosphatidylserine [54] on the platelet surface via flow cytometry. However, this method is not routine laboratory work. Platelet–leukocyte aggregates are often detected to show platelet activation [55], but this method is not quantitative (Table 3). Microparticles with tissue factor (TF) from platelets or vessels have been reported to be increased in patients with thrombosis [56], but this method is still being researched. Although, the β-thromboglobulin (β-TG), platelet factor 4 (PF4), and P-selectin are considered biomarkers of platelet activation, their diagnostic specificity for thrombosis due to platelet activation is not high, and their clinical laboratory use is inconvenient [57].

Soluble platelet membrane glycoprotein VI (sGPVI) and soluble C-type lectin-like receptor 2 (sCLEC-2) have been reported as new biomarkers for platelet activation [16,17,57]. Both sGPVI and sCLEC-2 were significantly elevated in patients with TMA and DIC [17,51]. Elevated sCLEC-2 levels were also reported in patients with ACS [18], ACI [58] and COVID-19 [46]. Specifically, the sCLEC-2/platelet ratio is useful for evaluating the severity of COVID-19. Furthermore, the plasma sCLEC-2 levels in patients with the mild stage of COVID-19 were similar to those in patients with other pneumonias, suggesting that the activation of platelets may occur in the early stage of COVID-19 without symptoms of microangiopathy [36]. Activated platelets in patients with COVID-19 may release large amounts of sCLEC-2 into the blood before causing severe microangiopathy. Although many reports have demonstrated decreased ADAMTS13 activity and increased von Willebrand factor (VWF) in patients with COVID-19 [59,60], ADAMTS-13 activity was not less than 10% in COVID-19 and the clinical usefulness of a mild decrease in ADAMTS-13 is not clear. Anti-PF 4 antibodies have often been reported in COVID-19 patients associated with thrombosis [61,62], suggesting that one of the thrombotic mechanisms in patients with COVID-19 is heparin-induced thrombocytopenia (HIT).

**Table 2 ijms-24-07975-t002:** Routine biomarkers for coagulopathy in COVID-19 infections and sepsis.

	COVID-19 Infection [14,15,52,54,55]			Sepsis Due to Bacterial Infection [37,38,57,59]		
	Cutoff Value	Sensitivity	Specificity	Cutoff Value	Sensitivity	Specificity
D-dimer	1.0–3.0 μg/mL	high	low	5–10 μg/mL	moderately high	moderately high
Platelet counts	16.0 × 10^10^/L	low	low	12.0 × 10^10^/L	moderately high	moderately high
PT-INR	1.20	low	low	1.20	moderately high	moderately high
Fibrinogen	increased	-	-	1.5 g/L	slightly high	high
Antithrombin	-	-	-	70%	moderately high	moderately high
WBC	decreased (at first)			markedly increased		
Hemoglobin	decreased (at severe or critical illness)			no change		

PT-INR, prothrombin time-internationalized ratio; WBC, white blood cells; COVID-19, coronavirus disease 2019.

**Table 3 ijms-24-07975-t003:** Examinations for platelet activation.

	Methods	Quantitative	Multiple Assay	Easy Assay	Specificity
Activated substance on platelet	flow cytometry	NA	NA	adequate	specific
Microparticles from platelet	flow cytometry, immunoassay	NA	NA	adequate	semispecific
Platelet–leukocyte aggregates	flow cytometry, microscopy	NA	NA	adequate	specific
β-TG, platelet factor 4 (PF4)	ELISA	PA	PA	NA	specific
P-secretin	ELISA	adequate	adequate	adequate	semispecific
GP-VI,	ELISA	adequate	adequate	adequate	specific
sCLEC-2	CLEIA	adequate	adequate	SA	specific

β-TG, β-thromboglobulin; PF4, platelet factor 4; NA, not adequate; PA, partially adequate; SA, strongly adequate; sGPVI, soluble platelet membrane glycoprotein VI; sCLEC-2, soluble C-type lectin-like receptor 2; ELISA, enzyme-linked immunosorbent assay, CLEIA, chemiluminescent enzyme immunoassay.

### 5.3. Hypofibrinolysis and Vascular Endothelial Cell Injury Markers

Increased fibrinogen [63] and plasminogen activator inhibitor-1 (PAI-I) levels [64], slightly increased D-dimer levels [65], and viscoelastic whole blood coagulation testing with and without tissue plasminogen activator [66,67] suggested a hypercoagulable and hypofibrinolytic state in patients with COVID-19. Most studies that reported hypofibrinolysis in patients with COVID-19 [66,67,68] used thromboelastography (TEG), and conducting statistical analyses for hypofibrinolysis proved difficult. Therefore, the hypofibrinolytic state in COVID-19 has not yet been sufficiently evaluated. Although it has been emphasized that D-dimer levels are increased in COVID-19 patients with severe or critical illness [36], the increase in D-dimer values in patients with COVID-19 has been shown to be significantly lower than that in other pneumonia patients [69]. Organ failure is worse in advanced COVID-19 patients, so vascular endothelial cell injury markers such as soluble thrombomodulin (sTM), VWF and PAI-I are high, while AT levels are low, suggesting that hypofibrinolysis may be related to organ failure and vascular endothelial cell injury.

### 5.4. Inflammatory Marker

Increased values for the white blood cell count, C-reactive protein (CRP) level [8], procalcitonin level [70], presepsin level [71], C5b-9 and C5a levels [44], and levels of inflammatory cytokines, such as tumor necrosis factor α, interleukin-1, interleukin-2, interleukin-6, interleukin-10 and interferon γ, were reported in patients with severe COVID-19; elevation in these inflammatory factors can lead to cytokine storm [72]. As procalcitonin is a biomarker for bacterial infection, presepsin may be more useful for diagnosing COVID-19 than procalcitonin [73]. These inflammatory mediators can further cause hypercoagulability, platelet activation, hypofibrinolysis, and vascular endothelial cell injuries by activating leukocytes, vascular endothelial cells and platelets.

## 6. Mechanisms for Thrombosis in COVID-19 and Sepsis

Several mechanisms for thrombosis underlying the worsening of the condition of COVID-19 patients, such as old age, long time-bed rest and comorbidities [23,28], inflammation and cytokine storms [12], vascular endothelial injuries [74], primary pulmonary thrombosis [75], hypoventilation, a hypercoagulable state (including activation of the TF pathway) [74], neutrophil extracellular traps (NETs) [76], hypofibrinolysis [66] and platelet activation [60], have been proposed. The mechanism underlying thrombosis in COVID-19 (Figure 3) and in bacterial infection (Figure 4) is shown.

### 6.1. Platelet Activation

Severe acute respiratory syndrome coronavirus-2 (SARS-CoV-2) binds to the CD-147 receptor of platelets [77]. Early and intense platelet activation, which was reproduced in vitro by stimulating platelets with SARS-CoV-2 depending on the CD147 receptor, has been reported [53]. Platelet activation and platelet–monocyte aggregate formation trigger TF expression in patients with severe COVID-19 [78]. SARS-CoV-2-induced platelet activation may participate in thrombus formation and inflammatory responses in COVID-19 patients. The early accumulation of extracellular vesicles with the soluble P-selectin and high mobility group box 1 (HMGB-1) protein which platelets release, was shown to predict worse clinical outcomes [53]. The plasma sCLEC-2 levels in patients with COVID-19 were significantly higher than those in patients with other infections and reflected the progression of the severity of COVID-19 although these levels were significantly higher in patients with sepsis due to bacterial infection [46] (Figure 4).

Thrombocytopenia is often observed in COVID-19 patients with severe disease as well as in septic patients with DIC [79]. The sCLEC-2/platelet ratio was significantly higher in COVID-19 patients with severe and critical illness than in those with mild illness, suggesting that one of the causes of thrombocytopenia might be consumption due to microthrombi formation, suggesting that COVID-19 has microangiopathy as well as DIC or TMA. Low-dose aspirin was reported to be useful for managing COVID-19 [80].

### 6.2. Hypercoagulable State

Marked activation of leukocytes and the overexpression of TF are considered some of the most important causes of thrombosis or DIC due to bacterial infection [14,79]. Markedly increased values of white blood cell count, plasma TF, and TF messenger RNA levels in white blood cells have been reported in patients with sepsis [81]. Increased levels of inflammatory cytokines, fibrin-related products (e.g., D-dimer), vascular endothelial cell injury markers (e.g., thrombomodulin [TM]), and PAI-I and decreased antithrombin, thrombocytopenia, and a prolonged PT were also reported in septic patients [14,38,82]. Such cases of sepsis are frequently associated with DIC [38]. Severe septic patients with elevated sCLEC-2 levels may also have microangiopathy [46]. There are some differences in the mechanism underlying thrombocytopenia between DIC and TMA [79], although thrombocytopenia in both diseases is caused by platelet consumption. Regarding COVID -19 (Figure 3), leukocyte counts are generally decreased early in COVID-19 [8], suggesting that activated platelets and injured vascular endothelial cells may play an important role in the onset of thrombosis through CD-147 [53]. However, CWA-APTT and CWA-sTF/FIXa showed hypercoagulability in patients with COVID-19 [41] suggesting that thrombin burst (Figure 2) [83] which is enhanced by activated platelets, causes hypercoagulability in this state.

### 6.3. Hypofibrinolysis

Although both severe COVID-19 and bacterial infections may have similar microangiopathy, complications with VTE are frequent in patients with COVID-19 but are not frequent in septic patients. DIC is caused by hypercoagulation and hyperfibrinolysis [38], most microthrombi in the microvasculature may dissolve promptly in DIC, whereas microthrombi in COVID-19 may develop into venous thrombosis. Vascular injuries are observed in COVID-19 [74], suggesting that elevated PAI-I may inhibit fibrinolysis [84] (Table 4 and Table 5). In addition, the sCLEC-2/D-dimer ratio in patients with COVID-19 was significantly higher than that in patients with other infections [69], suggesting that hypercoagulable and hypofibrinolysis states are more predominant in patients with COVID-19 than in other pneumonia patients. Markedly increased TF and D-dimer levels are observed in cases of bacterial infection [38,81], suggesting that hypercoagulable and hyperfibrinolytic states exist in severe bacterial infections. Regarding COVID-19 infection, the sCLEC-2/D-dimer ratio in cases with critical illness was significantly lower than that in cases with mild illness, suggesting that most patients with early-stage COVID-19 infection show only platelet activation with hypofibrinolysis, and that severe COVID-19 causes even further hypercoagulability [69] through a thrombin burst induced by platelet activation (Figure 4).

## 7. Treatment and Prophylaxis for Thrombosis

Although antithrombotic agents such as heparin reduce the risk of thromboembolism in severely ill patients, there are a few recommendations for patients with COVID-19 in the ISTH guidelines on antithrombotic treatment [85]. Among non-critically ill patients hospitalized for COVID-19, there is a strong recommendation for the use of prophylactic doses of low molecular weight heparin (LMWH) or unfractionated heparin (UFH) and for select patients in this group; the use of therapeutic doses LMWH/UFH is preferred over prophylactic doses, but without the addition of an antiplatelet agent. There are weak recommendations for adding an antiplatelet agent to prophylactic LMWH/UFH in select critically ill patients and prophylactic rivaroxaban for select patients after discharge. A recent review of RCTs [86] in critically ill patients demonstrated that a therapeutic dose of anticoagulation does not improve outcomes and results in more bleeding than a prophylactic dose of anticoagulant in these patients. Trials in noncritically ill hospitalized patients showed that anticoagulation at a therapeutic dose with a heparin formulation might improve clinical outcomes. Anticoagulation with a direct oral anticoagulant may improve outcomes of posthospital discharge; the results of a large RCT that is currently in progress are awaited [87]. There is not sufficient evidence that therapeutic anticoagulant can be recommended in critically ill patients at the present time.

## 8. Conclusions

A hypercoagulable state, platelet activation (observed as the marked elevation of sCLEC-2), and hypofibrinolysis due to vascular injuries are observed in patients with COVID-19, suggesting that SARS-CoV-2 may cause thrombogenicity via a mechanism different from that involved in bacterial infection.

## Figures and Tables

**Figure 1 ijms-24-07975-f001:**
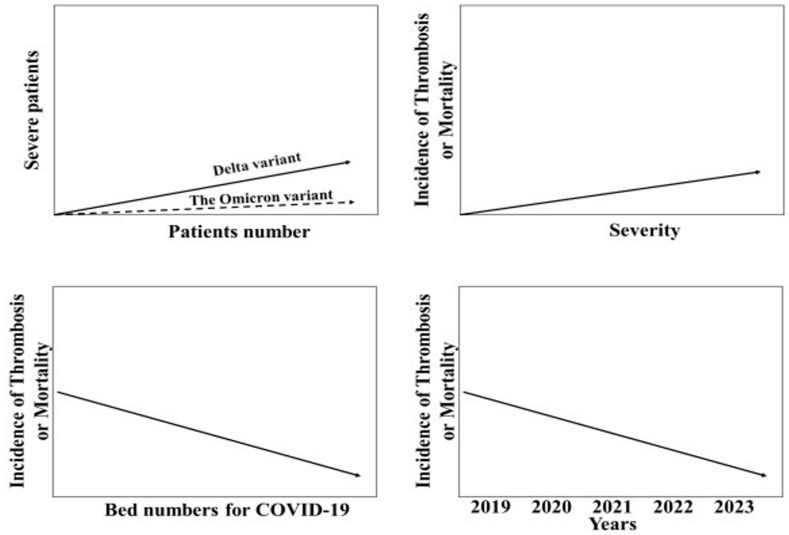
Variety of severity and complications of thrombosis in COVID-19.

**Figure 2 ijms-24-07975-f002:**
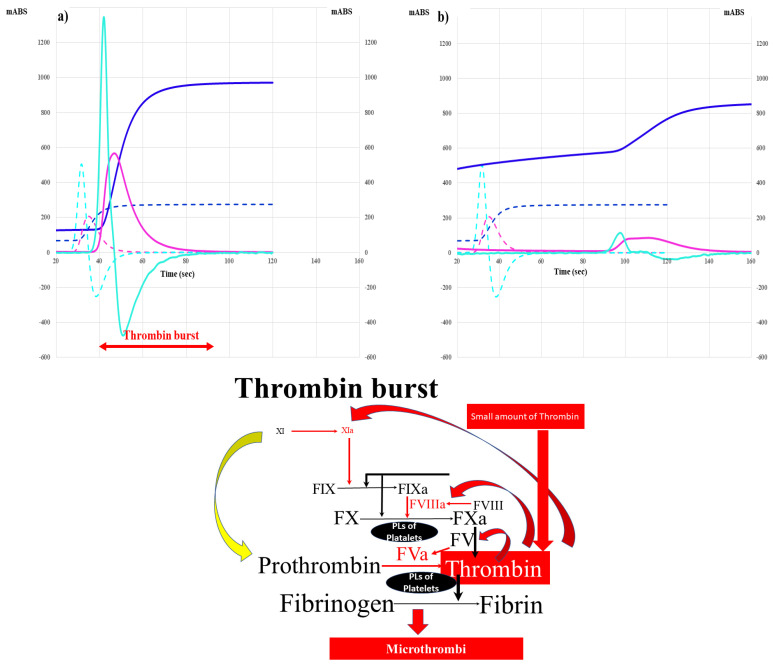
Difference in the CWA-APTT between COVID-19 coagulopathy with thrombin burst (**a**) and overt-DIC (**b**). CWA, clot waveform analysis; APTT, activated partial thromboplastin time; HV, healthy volunteer; COVID-19, coronavirus disease 2019; DIC, disseminated intravascular coagulation; navy line, fibrin formation curve; pink line, 1st derivative curve (velocity); light blue, 2nd derivative curve (acceleration); solid line, patient; dotted line, HV. A significant reduction in the peak height suggests bleeding, and a significant increase in the peak height suggest hypercoagulability and thrombotic risk. FX, activated FX; PLs, phospholipids; FVIIIa, activated FVIII, FVa, activated FV; FIXa, activated FIX; FXIa, activated FXI. A schematic illustration of thrombin burst in hypercoagulability with COVID-19.

**Figure 3 ijms-24-07975-f003:**
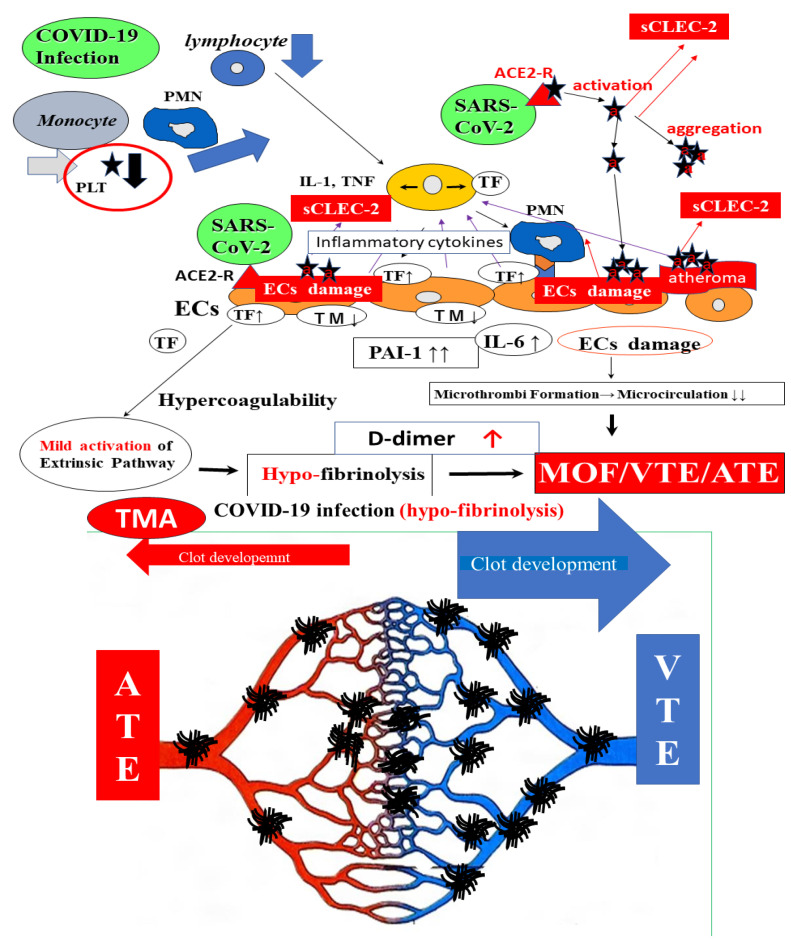
The mechanism underlying thrombosis in COVID-19. MΦ, macrophage; PMN, polymorphonuclear cell; NETS, neutrophil extracellular traps; ECs, endothelial cells; IFN, interferon; LPS, lipopolysaccharide; TF, tissue factor; IL, interleukin; TNF, tumor necrosis factor; TM, thrombomodulin; PAI-I, plasminogen activator inhibitor-1; AT-III, antithrombin; MOF, multiple organ failure; VTE, venous thromboembolism; DIC, disseminated intravascular coagulation; a★, activated platelet; PLT, platelet; COVID-19, coronavirus disease 2019; SARS-CoV-2, severe acute respiratory syndrome related coronavirus-2; sCLEC-2, soluble C-type lectin-like receptor 2; TMA, thrombotic microangiopathy; VTE, venous thromboembolism; ATE, arterial thromboembolism.

**Figure 4 ijms-24-07975-f004:**
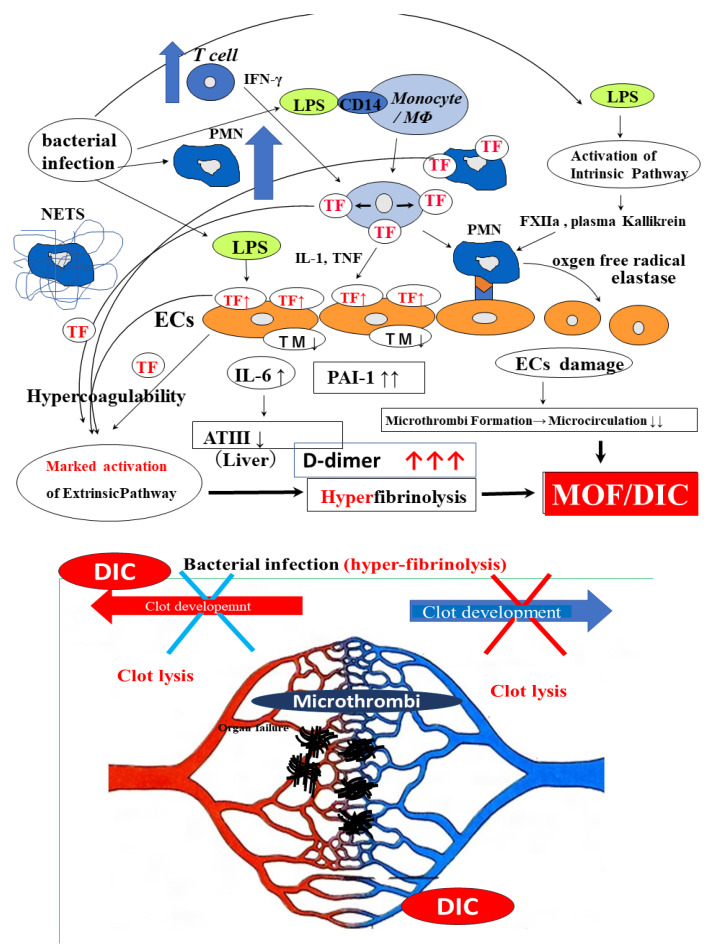
The mechanism underlying thrombosis in bacterial infection. MΦ, macrophage; PMN, polymorphonuclear cell; NETS, neutrophil extracellular traps; ECs, endothelial cells; IFN, interferon; LPS, lipopolysaccharide; TF, tissue factor; IL, interleukin; TNF, tumor necrosis factor; TM, thrombomodulin; PAI-I, plasminogen activator inhibitor-1; AT-III, antithrombin; MOF, multiple organ failure; VTE, venous thromboembolism; DIC, disseminated intravascular coagulation.

**Table 4 ijms-24-07975-t004:** Differences and similarities between COVID-19 and severe sepsis bacterial infections.

	COVID-19 Infection	Severe Sepsis Due to Bacterial Infections
Activation of platelets	+++++	+++
Activation of leukocytes	+	+++++
Tissue factor generation	++++	+++++
Cytokine generation	+++++	++++
Lung injury	+++++	+++
Organ failure excluding lung	+	+++
Development of atheroma	+++	+
Development of venous thrombosis	+++++	++
Fibrinolysis	+	+++++

**Table 5 ijms-24-07975-t005:** Differences and similarities between COVID-19 and severe sepsis bacterial infections.

	COVID-19 Infection	Severe Sepsis Due to Bacterial Infections
Venous thromboembolism	frequent	not frequent
Arterial thrombosis	relatively frequent	not frequent
Mortality rate	approximately 2%	20–45% in severe sepsis
Incidence of infection	markedly high	relatively high
Death number	markedly high	relatively high
Microangiopathy	positive	positive
Coagulation	mild or hypercoagulable states	hypercoagulable state
Fibrinolysis	hypofibrinolytic state	hyperfibrinolytic state
DIC	not frequent	frequent

DIC, disseminated intravascular coagulation; COVID-19, coronavirus disease 2019.

## Data Availability

Yes, available in References.

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
