# Peer review of "Thrombotic Mechanism Involving Platelet Activation, Hypercoagulability and Hypofibrinolysis in Coronavirus Disease 2019"

_ijms, 2023, doi:10.3390/ijms24097975_

Round 1

Reviewer 1 Report

The authors Wada H et al. have submitted a nice review regarding thrombotic mechanisms involved in coronovirus disease 2019. 

however, some aspects need to be improved: 

1. In figure 3, the figures start with COVID associated mechanisms followed by bacterial infection on the right side; the Figure description below in contrary starts with bacterial infection followed by COVID-19 related mechanisms. In addition, the figures do not show a) or b). The differences between both infection situation should be made more clear. 

2. The reference list is quite long and should be shortened. 

3. It would be of interest to further include a section to discuss possible therapeutic options, based on the available literature, to prevent COVID-10 associated venous or arterial thrombosis.    

just minor corrections required

Author Response

Thank you for your valuable review. We have fully revised our manuscript in accordance with your comments. Yellow color showed our revised words.

The authors Wada H et al. have submitted a nice review regarding thrombotic mechanisms involved in coronovirus disease 2019. however, some aspects need to be improved: 

Comment 1.                                                                                                                                   1. In figure 3, the figures start with COVID associated mechanisms followed by bacterial infection on the right side; the Figure description below in contrary starts with bacterial infection followed by COVID-19 related mechanisms. In addition, the figures do not show a) or b). The differences between both infection situation should be made more clear. 

Response 1.  Figure 3 and its legend have been revised for clarity.

Comment2.                                                                                                                               The reference list is quite long and should be shortened. 

Response 2. The references have been fully revised.

Comment3.                                                                                                              3. It would be of interest to further include a section to discuss possible therapeutic options, based on the available literature, to prevent COVID-10 associated venous or arterial hrombosis. 

Response 3. A section concerning the treatment of and prophylaxis for thrombosis section has been added.

Reviewer 2 Report

In the presented review, Wada et al compared thrombotic complications in septic and COVID-19 patients and indicated that that hypercoagulable and hyperfibrinolytic states are predominant in bacterial infections, whereas hypercoagulable and hypofibrinolytic states with platelet activation are predominant in COVID-19. The authors presented numerous data concerning mechanisms of thrombosis in septic and COVID-19 patients and indicated differences underling these processes in two type of infection. The review is good written and important for understandingthe mechanisms of numerous thrombotic complications during infection.

I have only a small suggestions to the authors.

1.       In Fig. 2 A and B is not indicated. On the p. 9, l 303. “Regarding COVID -19 (Figure 2b), leukocyte counts are generally decreased early in COVID-19” Fig. 2b is mentioned, however there are no Fig. 2b and this fig. is not associated with leukocite.

2.       Similarly for Fig. 3. a and b are not indicated. In the p. 9 “with sepsis due to bacterial infection [52] (Figure 3a)”. Here should be Fig 3b not 3a, in Fig 3a COVID-19 is shown.

3.       The quality of Figs 2 and 4 are not satisfactory and they should be improved.   

Author Response

Thank you for your valuable review. We have fully revised our manuscript in accordance with your comments. Yellow color showed our revised words.

In the presented review, Wada et al compared thrombotic complications in septic and COVID-19 patients and indicated that that hypercoagulable and hyperfibrinolytic states are predominant in bacterial infections, whereas hypercoagulable and hypofibrinolytic states with platelet activation are predominant in COVID-19. The authors presented numerous data concerning mechanisms of thrombosis in septic and COVID-19 patients and indicated differences underling these processes in two type of infection. The review is good written and important for understandingthe mechanisms of numerous thrombotic complications during infection.

I have only a small suggestions to the authors.

Comment 1.                                                                                                                                           

In Fig. 2 A and B is not indicated. On the p. 9, l 303. “Regarding COVID -19 (Figure 2b), leukocyte counts are generally decreased early in COVID-19” Fig. 2b is mentioned, however there are no Fig. 2b and this fig. is not associated with leukocite.

Response 1. Figure 2b has been changed to Figure 3a.

Comment 2.                                                                                                                                                   

Similarly for Fig. 3. a and b are not indicated. In the p. 9 “with sepsis due to bacterial infection [52] (Figure 3a)”. Here should be Fig 3b not 3a, in Fig 3a COVID-19 is shown.

Response 2. These sentences and the legend of Figure 3 have been revised.

Comment 3.                                                                                                                                                   

 The quality of Figs 2 and 4 are not satisfactory and they should be improved.   

Response 3. Figures 2 and 4 have been revised.

Round 2

Reviewer 1 Report

The authors have improved their manuscrip based on the recommendations given by the reviewers. Some minor points need further improvement. 

1. The authors have now included a paragraph (page 10) dealing with the preventive and therapeutic options to prevent or treat COVID-19 associated arterial or venous thrombotic events. 

Please mention the sudies performed regarding these points supporting the recommendation to administer UFH, LMWH or OAK for prevention or treatment. Did these studies provide evidence based recommendations ? 

2.  in line 335, is it correct to mention and or better or regarding the treatment options. 

3. Figures 3 and 4 are still full of information and difficult to read. In figure 3 a and b a more clear separartion between the two figures would be helpful, although the text should still be readable.

In addition, the table below figure 3 should be improved not expanding the letters in the present way.   

The authors have improved their manuscrip based on the recommendations given by the reviewers. Some minor points need further improvement. 

1. The authors have now included a paragraph (page 10) dealing with the preventive and therapeutic options to prevent or treat COVID-19 associated arterial or venous thrombotic events. 

Please mention the sudies performed regarding these points supporting the recommendation to administer UFH, LMWH or OAK for prevention or treatment. Did these studies provide evidence based recommendations ? 

2.  in line 335, is it correct to mention and or better or regarding the treatment options. 

3. Figures 3 and 4 are still full of information and difficult to read. In figure 3 a and b a more clear separartion between the two figures would be helpful, although the text should still be readable.

In addition, the table below figure 3 should be improved not expanding the letters in the present way.   

Author Response

Thank you for your valuable comments. We have fully revised our manuscript in accordance with the comments from reviewer 1.

Comments and Suggestions for Authors

The authors have improved their manuscrip based on the recommendations given by the reviewers. Some minor points need further improvement.

Comments 1 and 2

  1. The authors have now included a paragraph (page 10) dealing with the preventive and therapeutic options to prevent or treat COVID-19 associated arterial or venous thrombotic events.

Please mention the sudies performed regarding these points supporting the recommendation to administer UFH, LMWH or OAK for prevention or treatment. Did these studies provide evidence based recommendations?

  1. in line 335, is it correct to mention andor better orregarding the treatment options. 

Responses 1 and 2. We have fully revised this section in accordance with the comments from reviewer 1.

Comments3

  1. 3. Figures 3 and 4 are still full of information and difficult to read. In figure 3 a and b a more clear separartion between the two figures would be helpful, although the text should still be readable.

In addition, the table below figure 3 should be improved not expanding the letters in the present way.   

Responses 3. Figures 3 and 4 have been revised and combined, and have been separated into Figures 3 (COVID-19) and Figures 4 (bacterial infections). A schematic illustration of thrombin burst  has been added in Figure 2.